# Effect of *Ephestia kuehniella* Eggs on Development and Transcriptome of the Ladybird Beetle *Propylea japonica*

**DOI:** 10.3390/insects15060407

**Published:** 2024-06-02

**Authors:** Guannan Li, Pei-Tao Chen, Mei-Lan Chen, Tuo-Yan Chen, Yu-Hao Huang, Xin Lü, Hao-Sen Li, Hong Pang

**Affiliations:** 1State Key Laboratory of Biocontrol, School of Ecology, Sun Yat-sen University, Shenzhen 518000, China; lign6@mail.sysu.edu.cn (G.L.); chenpt3@mail2.sysu.edu.cn (P.-T.C.); chenmlan3@mail2.sysu.edu.cn (M.-L.C.); chenty58@mail2.sysu.edu.cn (T.-Y.C.); huangyh45@mail2.sysu.edu.cn (Y.-H.H.); 2Guangdong Key Laboratory of Animal Conservation and Resource Utilization, Guangdong Public Laboratory of Wild Animal Conservation and Utilization, Institute of Zoology, Guangdong Academy of Sciences, Guangzhou 510275, China; greenhopelv@163.com

**Keywords:** flour moth eggs, predatory ladybird, biological control, life history, transcriptome

## Abstract

**Simple Summary:**

Improving augmentative biological control relies on the development of a cost-effective and readily available diet for rearing natural enemy insects. This study highlights the effectiveness of *Ephestia kuehniella* eggs in promoting the development of the predatory ladybird beetle *Propylea japonica*. It represents the first study in assessing the advantages of *E. kuehniella* eggs as dietary sources for ladybirds by examining transcriptional regulation.

**Abstract:**

The eggs of the Mediterranean flour moth, *Ephestia kuehniella*, are frequently utilized as alternative diets and have demonstrated promising outcomes when consumed by various insects. Nonetheless, the specific reasons for their effectiveness remain unclear. In our study, we assessed the developmental performance of the ladybird *Propylea japonica* when fed *E. kuehniella* eggs, alongside 12 factitious prey or artificial diets. Our findings revealed that ladybirds fed *E. kuehniella* eggs displayed a performance comparable to those fed the natural prey *Megoura crassicauda*. Transcriptome profiling of larvae raised on *E. kuehniella* eggs and *M. crassicauda* revealed that genes upregulated in the former group were enriched in metabolic pathways associated with carbohydrates, lipids, and other essential nutrients. This suggests that *E. kuehniella* eggs may have a higher nutrient content compared to natural prey. Furthermore, a notable downregulation in the expression of immune effector genes, such as *Attacin* and *Coleoptericin*, was observed, which might be attributed to the lower microbial content in *E. kuehniella* eggs compared to *M. crassicauda*. We suggest that the difference between *E. kuehniella* eggs and *M. crassicauda* as food sources for *P. japonica* lies in their nutrient and microbial contents. These findings provide valuable insights for the advancement of innovative artificial breeding systems for natural enemies.

## 1. Introduction

Augmentative biological control stands as an important method in agricultural pest management, involving the release of large quantities of natural enemies into the field [1,2]. The production of these natural enemies relies on a three-tiered artificial rearing system encompassing predators, prey, and the prey’s host plants. Nonetheless, challenges such as discontinuity and high rearing facility costs can impede this method’s efficacy [3,4]. Thus, the development of a cost-effective and easily accessible artificial diet is crucial for enhancing biological control strategies.

The eggs of the Mediterranean flour moth, *Ephestia kuehniella* Zeller, 1879 (Lepidoptera, Pyralidae), are widely used as a factitious diet [5]. They offer the advantage of being easily obtainable and preservable. These eggs are reported to contain higher levels of nutrients, with three times more amino acids and lipids than aphids like *Acyrthosiphon pisum* (Harris, 1776) (Hemiptera, Aphididae) [5]. These eggs have been successfully utilized to feed various predatory ladybirds such as *Adalia bipunctata* (Linnaeus, 1758) (Coleoptera, Coccinellidae), *Harmonia axyridis* (Pallas, 1773) (Coleoptera, Coccinellidae), and *Cryptolaemus montrouzieri* Mulsant, 1853 (Coleoptera, Coccinellidae) [4,6,7,8,9,10]. They have been shown to enhance growth and developmental rates compared to natural prey options [5,7,10] and also serve as hosts for parasitoid natural enemies like *Trichogramma* (Hymenoptera, Trichogrammatidae) [11]. However, the suitability of *E. kuehniella* eggs varies among different ladybirds, as they are not suitable for rearing *Coccinella septempunctata* (Linnaeus, 1758) (Coleoptera, Coccinellidae) [6,12]. The specific reasons for the effectiveness of *E. kuehniella* eggs as ladybird diets are still largely unknown.

Predatory ladybirds are commonly used as natural enemies in augmentative biological control due to their pest-feeding abilities [4]. *Propylea japonica* (Thunberg, 1781) (Coleoptera, Coccinellidae), a native of Asia, predominantly preys on aphids and various agricultural pests. Known for its adaptability to diverse diet regimes and temperature ranges, *P. japonica* are widely used in augmentative biological control [13]. This study aimed to assess the suitability of *E. kuehniella* eggs as a diet for *P. japonica* and explore the differences between *E. kuehniella* eggs and natural prey as food sources for this ladybird. We compared the life history traits of *P. japonica* when fed on 14 diets, which included natural prey like *Megoura crassicauda* Mordvilko, 1919 (Hemiptera, Aphididae) aphids, factitious prey such as *E. kuehniella* eggs, and artificial diets. We also performed transcriptome profiling to identify differentially expressed genes (DEGs) in *P. japonica* in response to feeding on *E. kuehniella* eggs.

## 2. Materials and Methods

### 2.1. Preparing of the Ladybird and Diets

The ladybird *P. japonica* in the experiment was collected from Baiyun Mountain, Guangzhou, China (113°17′ E, 23°11′ N) between 2014 and 2018. Before the experiment, *P. japonica* eggs were harvested from the cages using plastic straws (0.5 cm in diameter, 3 cm in length) and transferred to plastic Petri dishes (15 cm in diameter, 1.5 cm in height). The hatched larvae from these eggs were then used in the study. A total of 14 diet treatments (Table 1) were assessed for the rearing *P. japonica*. Nine diet materials were formulated into a culture medium based on a previously published recipe [14], comprising 2.5 g of sucrose, 1.5 g of a protein source, 0.8 g of agarose, and 10 mL of distilled water. The eggs of *E. kuehniella* (FLO) and *Spodoptera frugiperda* (FAL) were stored in a freezer at −20 °C and thawed at 4 °C before use. 

### 2.2. Feeding Experiments of P. japonica on Different Diets

The feeding experiments were performed from March, 2017 to December, 2023. Newly emerged male and female adults were paired in plastic Petri dishes (9 cm in diameter, 1.5 cm in height), and the eggs they laid were collected. The newly hatched larvae were individually placed into plastic Petri dishes (3 cm in diameter, 1 cm in height) and divided into 14 groups with varying diets. Each dish was equipped with a moist cotton to ensure water availability. Throughout the experiment, the total developmental time, adult weight, and survival rate were documented daily, with daily replenishment of food. Adults were weighed on the day of emergence. The experiment included a total of 14 treatments and 24 batches (Table 1). To evaluate the possible batch effect of the main treatments APH and FLO, three batches were conducted for each. For the other treatments, one to three batches were conducted. Each batch comprised 22–99 tested individuals (Table 1). All feeding experiments were performed in climatic chambers at 25 (±1) °C, 75 (±5) % RH under a 16:8 (L: D) photoperiod. Statistical analyses for comparing life history traits among different treatments and batches were carried out using R software v4.2.3. The Shapiro–Wilk normality test and Bartlett’s test were employed to assess the normality and homogeneity of variance for the developmental time and adult weight data. Data conforming to these assumptions underwent one-way ANOVA analysis. In cases where a significant difference was observed, Tukey’s Honestly Significant Difference (HSD) test was utilized for post hoc comparisons. Alternatively, the Wilcoxon signed-rank test was applied. A significance level of *p* < 0.05 was adopted for all statistical tests.

### 2.3. Transcriptome Analysis

Fourth-instar larvae (<24 h after molting) of *P. japonica* from the above batches APH01 and FLO02 were randomly collected for transcriptome analysis. An individual larva was used as a single biological replicate, and three and two biological replicates were set for APH01 and FLO02, respectively. The total RNA of each individual was extracted using TRIzol reagent (CWBIO, Beijing, China). RNA quality and quantity were determined using a Nanodrop 1000 spectrophotometer (Thermo Fisher Scientific, Wilmington, NC, USA). Only RNA samples with a 260/280 ratio from 1.8 to 2.0, a 260/230 ratio from 2.0 to 2.5, and an RNA integrity number (RIN) > 8.0 were used for sequencing. Sequencing was performed on the Illumina HiSeq 2500 platform. Adaptors and low-quality sequences were removed using the default settings for Trimmomatic v0.36 [15]. The clean reads were then mapped to the published genome of *P. japonica* [16] using HISAT2 v2.2.1 [17]. Abundance estimation was performed using StringTie v2.1.4 [18] based on genome annotation. Differential expression analysis between treatments was performed using DESeq2 [19] according to the standard workflow, with a log2(fold change) (log2FC) value > 1 or <−1 and an adjusted *p* value (Q value) < 0.05 used as the criteria for defining differentially expressed genes (DEGs). Gene Ontology (GO) and Kyoto Encyclopedia of Genes and Genomes (KEGG) pathway annotations were performed using eggnog-mapper v2 (eggNOG database v5.0) [20,21]. Functional enrichment of DEGs was performed in the R package clusterProfiler [22]. Visualization of GO enrichment was performed using GO-Figure! [23].

## 3. Results

### 3.1. The Analysis of Life History Traits of P. japonica on Different Diet

In the analysis of life history traits, 9 out of the 14 treatments (APH, FLO, PUP, HON, WHI, POL, BRI, FAL, and YEL) led to individuals reaching the adult stage (Figure 1A). The survival rates of APH were generally lower than those of FLO (Figure 1A). Treatments WHI, POL, BRI, FAL, and YEL had less than 25% of the tested individuals reaching adulthood (Figure 1A), and their developmental time and adult weight data were excluded from further analysis. Figure 1B integrated the parameters of survival rate, developmental time, and adult weight, demonstrating that APH and FLO exhibited superior performance compared to PUP and HON. None of the life history traits were significantly different between the APH and FLO treatments (Table 2). In addition, the performance of APH and FLO exhibited variability among batches. For example, the development time of a batch in FLO (FLO02) was significantly shorter than not only batches in APH (APH02–03) but also FLO (FLO01) (*p* < 0.05 in Wilcoxon signed-rank test) (Figure 1C). Similarly, the adult weight of a batch in APH (APH03) was significantly higher than batches in both APH (APH02) and FLO (FLO02–03) (*p* < 0.05 in Wilcoxon signed-rank test) (Figure 1D).

### 3.2. The Analysis of Transcriptome Profiling between APH and FLO

In the comparative analysis of transcriptome profiling between APH (as control) and FLO (as treatment), the samples from the treatment group displayed coefficient of determination (*r*^2^) values ranging from 0.82 to 0.93, with a mean of 0.88 (Table 3). A total of 479 differentially expressed genes were identified, with 345 genes upregulated and 134 genes downregulated. Specifically, immune effector genes such as *Attacin* and *Coleoptericin* were significantly downregulated (Figure 2). The results of the Gene Ontology (GO) enrichment analysis revealed that the significantly upregulated genes were mainly enriched in nutritional metabolic functions, including catalytic activity, transmembrane transporter activity in the Molecular Function (MF) category, and carbohydrate metabolic processes in the Biological Process (BP) category (Figure 3). Conversely, the downregulated genes were only enriched in two GO terms: 4-hydroxyphenylpyruvate dioxygenase activity and tyrosine catabolic process. The Kyoto Encyclopedia of Genes and Genomes (KEGG) enrichment analysis showed similar findings, with upregulated genes being enriched in pathways related to glycogen metabolism (Table 4).

## 4. Discussion

In the feeding experiment, most of the artificial diets resulted in a poorer performance of *P. japonica* than the natural prey pea aphids *M. crassicauda*. Some diets could only support a few ladybirds to mature into adults, such as whiteflies (WHI treatment) and the pollen of *Brassica campestris* (POL treatment). Other diets, like the pupae of the honeybee *Apis mellifera* (PUP treatment) and bee honey (HON treatment), could sustain over 50% of larvae to reach the adult stage, but at the cost of a significantly decreased larval development time and adult weight. Hence, none of these diets were deemed suitable for *P. japonica*. Among all the tested artificial diets, the natural prey pea aphids *M. crassicauda* (APH treatment) and *E. kuehniella* eggs (FLO treatment) exhibited the best performance for feeding *P. japonica*. Although there was variability in performance among batches in our experiments, the life history traits of *P. japonica* reared on both *E. kuehniella* eggs and aphids showed relatively similar patterns. This finding aligns with previous reports that identified *E. kuehniella* eggs as a viable alternative diet for the development and reproduction of *P. japonica* [8]. Therefore, *E. kuehniella* eggs are suitable for use as an alternative food source in the artificial feeding system of *P. japonica*.

By comparing the transcriptome of *P. japonica* fed on *E. kuehniella* eggs with that of aphids, we identified numerous upregulated metabolism-related genes that were enriched in several sugar metabolism pathways. This outcome parallels findings in previous studies on other ladybird species, indicating that changes in life history traits and transcriptome regulation may reflect variations in the nutritional composition of the *E. kuehniella* eggs [14,24]. Similar changes in life history traits and differentially expressed metabolism-related genes in response to varied nutritional compositions of diets have been observed in other insects. For example, nutrient-sensing and metabolic pathways in *A. mellifera* are activated by feeding on honey, which contains higher levels of proteins and amino acids, compared to sugar feeding [25]. These results underscore the importance of nutrient and biochemical composition in assessing the suitability of an artificial diet [26]. However, we did not test the nutritional content differences between *E. kuehniella* eggs and pea aphids *M. crassicauda*. Additionally, we cannot rule out the possibility that other factors also contributed to the differential expression of metabolism-related genes.

Aphids are known to harbor various symbiotic bacteria, such as *Buchnera aphidicola*, *Serratia symbiotica*, *Hamiltonella defensa*, *Regiella insecticola*, *Rickettsia*, *Rickettsiella*, *Spiroplasma*, *Wolbachia*, and *Arsenophonus* [27]. Some of these bacteria may have adverse effects on the natural enemies to protect their host [28,29]. These bacteria can change the gene expression in the natural enemies [30]. In contrast, *E. kuehniella* eggs contain fewer microbes that could elicit changes in gene expression in ladybirds. Previous studies have reported that the >99% microbiota composition of these eggs belonged to an intracellular bacterium, *Wolbachia* [24]. In this study, two immune effector genes of *P. japonica* were significantly downregulated when feeding on *E. kuehniella* eggs, suggesting that these genes may play a crucial role in eliminating bacteria from aphids. Therefore, feeding on *E. kuehniella* eggs may result in less immune stress, contributing to the superior performance of *P. japonica* feeding on this diet.

Our research highlights that *E. kuehniella* eggs are a suitable artificial diet for the mass-rearing of *P. japonica*, surpassing other tested alternative prey or artificial diets. Through the transcriptome profiling analysis, we suggest that the difference between *E. kuehniella* eggs and *M. crassicauda* as food sources for *P. japonica* lies in their nutrient and microbial contents. These findings also offer valuable insights for developing improved systems for the captive breeding of other natural enemies. However, the effects of feeding *E. kuehniella* eggs to multiple successive generations of *P. japonica* remain unclear and need further research for clarification.

## Figures and Tables

**Figure 1 insects-15-00407-f001:**
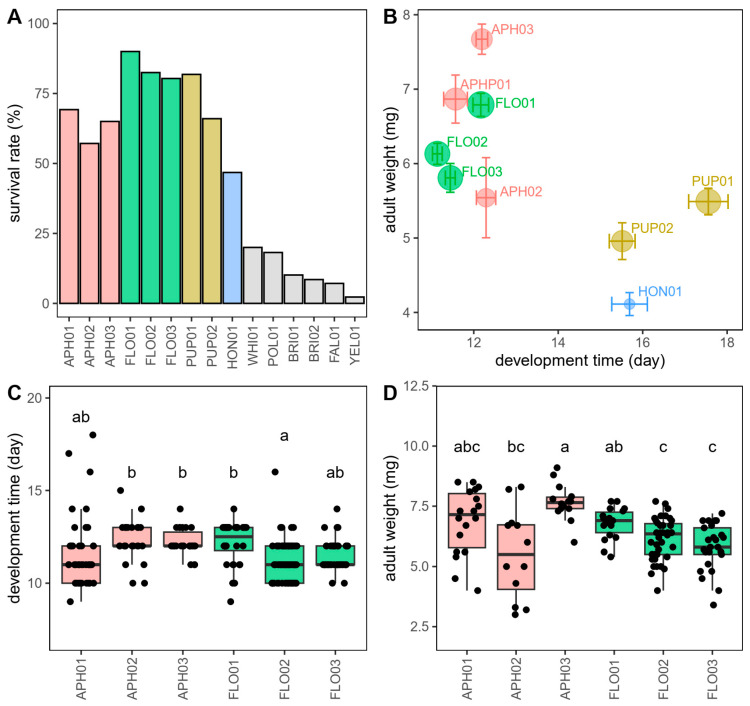
Effects of different diet treatments and batches on life history traits of *Propylea japonica*. (**A**) Survival rate of batches in APH, FLO, PUP, HON, WHI, POL, BRI, FAL, and YEL treatments. (**B**) Survival rate (point size), development time (X-axis), and adult weight (Y-axis) of batches in APH, FLO, PUP, and HON treatments. (**C**) Development time and (**D**) female adult weight of batches in APH and FLO treatments. Different colors indicate different diet treatments. Bars with the same letter are not significantly different (*p* ≥ 0.05).

**Figure 2 insects-15-00407-f002:**
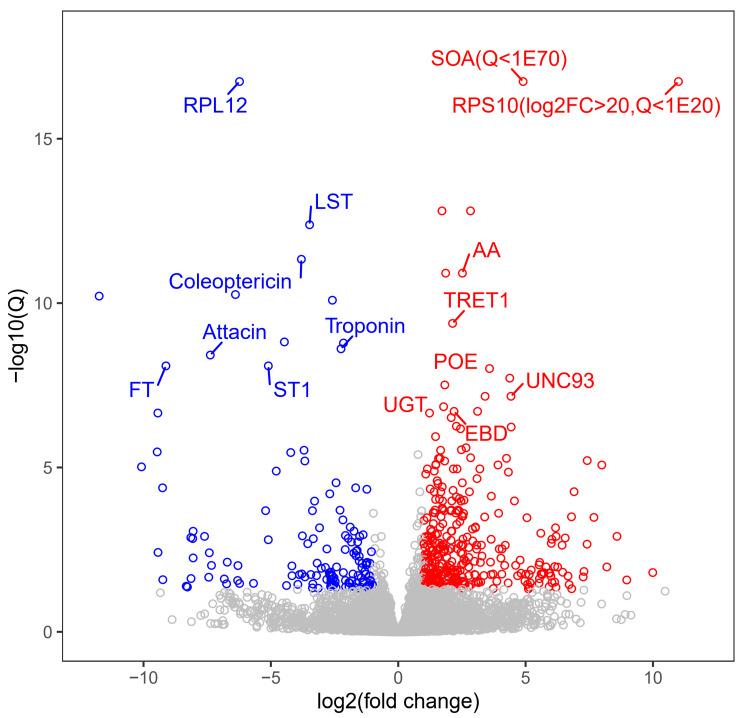
Volcano plot of the transcriptome comparison in *Propylea japonica* when feeding on *Ephestia kuehniella* eggs (FLO treatment) compared to pea aphids *Megoura crassicauda* (APH treatment). X-axis: log_2_ of the fold change values. Y-axis: −log_10_ of the adjusted *p* values (Q). Red: upregulated differentially expressed genes (DEGs). Blue: downregulated DEGs. SOA: Sterol O-acyltransferase 1, RPS10: 40S ribosomal protein S10, RPL12: 60S ribosomal protein L12, LST: Luciferin sulfotransferase, AA: Alpha-amylase, TRET1: Trehalose transporter 1, ST1: Sulfotransferase 1C2, FT: Formyl transferase, POE: Protein obstructor-E, UNC93: Ion channel regulatory protein, UGT: UDP-glycosyltransferase, EBD: Estradiol 17-beta-dehydrogenase 11. Larger absolute values on the X-axis and higher values on the Y-axis indicate greater differential expression of genes.

**Figure 3 insects-15-00407-f003:**
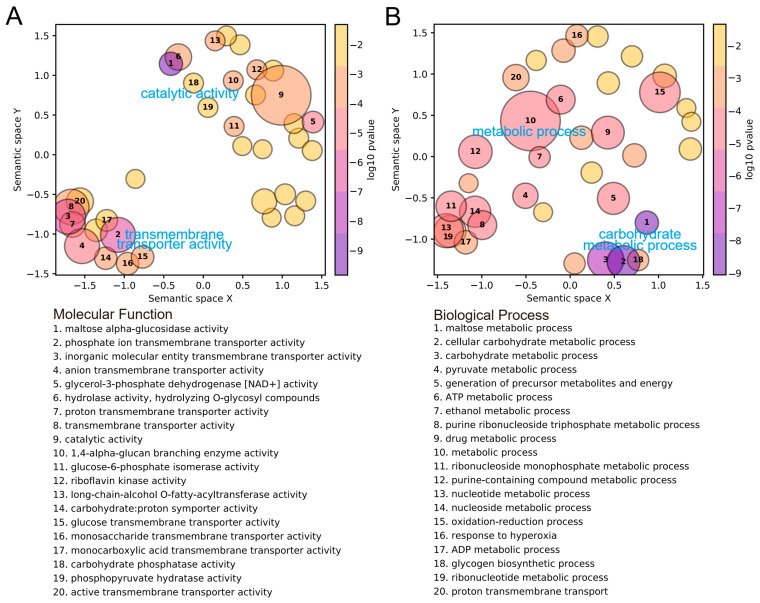
Gene Ontology (GO) enrichment analysis of differentially expressed genes in *Propylea japonica* when feeding on *Ephestia kuehniella* eggs (FLO treatment) compared to pea aphids *Megoura crassicauda* (APH treatment). Each bubble represents a cluster of similar GO terms summarized by a representative term reported in the legend and sorted by the average *p* values of the representative GO term across the module of (**A**) Molecular Function and (**B**) Biological Process. Bubble size indicates the number of GO terms in each cluster, and the color is the average *p* value of the representative GO term across the gene modules. Similar clusters plot closer to each other.

**Table 1 insects-15-00407-t001:** Information of the treatments, batches, number of individuals tested, and dates of the experiments conducted in this study.

Treatment	Batch	DIET MATERIAL	Process for Diet	Individual	Date
APH	APH01	Pea aphids *Megoura crassicauda*	Live prey	65	October 2021
APH	APH02	Pea aphids *Megoura crassicauda*	Live prey	42	Novermber 2018
APH	APH03	Pea aphids *Megoura crassicauda*	Live prey	40	September 2022
FLO	FLO01	Eggs of flour moth *Ephestia kuehniella*	Cold-stored	40	August 2021
FLO	FLO02	Eggs of flour moth *Ephestia kuehniella*	Cold-stored	97	October 2021
FLO	FLO03	Eggs of flour moth *Ephestia kuehniella*	Cold-stored	56	May 2021
PUP	PUP01	Pupae of honeybee *Apis mellifera*	Dry powder and solid medium	22	March 2017
PUP	PUP02	Pupae of honeybee *Apis mellifera*	Dry powder and solid medium	50	March 2017
HON	HON01	Bee honey	Solid medium	62	Novermber 2018
HON	HON02	Bee honey	Solid medium	22	March 2017
WHI	WHI01	Whitefies *Bemisia tabaci*	Live prey	50	Novermber 2018
POL	POL01	Pollen of Brassica campestris	Plant materials	99	Novermber 2023
BRI	BRI01	Cysts of brine shrimp *Artemia salina*	Solid medium	69	Novermber 2018
BRI	BRI02	Cysts of brine shrimp *Artemia salina*	Solid medium	47	March 2017
FAL	FAL01	Eggs of fall armyworm *Spodoptera frugiperda*	Cold-stored	28	December 2023
YEL	YEL01	Larvae of yellow mealworm *Tenebrio molitor*	Dry powder and solid medium	88	Novermber 2018
YEL	YEL02	Larvae of yellow mealworm *Tenebrio molitor*	Dry powder and solid medium	23	March 2017
YEL	YEL03	Larvae of yellow mealworm *Tenebrio molitor*	Dry powder and solid medium	29	March 2017
BLA	BLA01	Larvae of black soldier fly *Hermetia illucens*	Dry powder and solid medium	28	March 2017
BLA	BLA02	Larvae of black soldier fly *Hermetia illucens*	Dry powder and solid medium	30	May 2021
SIL	SIL01	Pupae of silkworm *Bombyx mori*	Dry powder and solid medium	74	March 2017
POR	POR01	Pork	Dry powder and solid medium	20	March 2017
LIV	LIV01	Pork liver	Dry powder and solid medium	18	March 2017
CHI	CHI01	Chicken egg	Solid medium	25	March 2017

**Table 2 insects-15-00407-t002:** Comparison of life history traits in *Propylea japonica* when feeding on *Ephestia kuehniella* eggs (FLO treatment) compared to pea aphids *Megoura crassicauda* (APH treatment).

	APH	FLO	Wilcoxon Signed-Rank Test
Survival rate (%)	63.792 ± 3.541	84.277 ± 2.926	Not significant
Development time (day)	12.017 ± 0.226	11.583 ± 0.305	Not significant
Female adult weight (mg)	6.693 ± 0.621	6.243 ± 0.289	Not significant
Male adult weight (mg)	5.484 ± 0.308	5.247 ± 0.307	Not significant

**Table 3 insects-15-00407-t003:** Coefficient of determination (*r*^2^) values of gene expression between the studied transcriptomes.

	APH	APH	APH	FLO	FLO
APH					
APH	0.890				
APH	0.910	0.922			
FLO	0.862	0.942	0.908		
FLO	0.820	0.821	0.874	0.889	

**Table 4 insects-15-00407-t004:** Kyoto Encyclopedia of Genes and Genomes (KEGG) enrichment analysis of differentially expressed genes in *Propylea japonica* when feeding on *Ephestia kuehniella* eggs (FLO treatment) compared to pea aphids *Megoura crassicauda* (APH treatment).

Regulation	ID	Descrption	Q Value	Ratio
up	ko00500	Metabolism: Starch and sucrose metabolism	7.30 × 10^−15^	16/59
up	ko01100	Metabolism: Metabolic pathways	2.44 × 10^−9^	69/2498
up	ko00052	Metabolism: Galactose metabolism	9.28 × 10^−7^	11/83
up	ko01200	Metabolism: Carbon metabolism	1.47 × 10^−4^	14/231
up	ko00010	Metabolism: Glycolysis/Gluconeogenesis	5.78 × 10^−4^	9/107
up	ko00830	Metabolism: Retinol metabolism	0.0069	9/153
up	ko00310	Metabolism: Lysine degradation	0.0069	8/121
up	ko00980	Metabolism: Metabolism of xenobiotics by cytochrome P450	0.0095	9/163
up	ko00053	Metabolism: Steroid hormone biosynthesis	0.0134	8/143
up	ko00650	Metabolism: Ascorbate and aldarate metabolism	0.0148	4/32
up	ko00982	Metabolism: Butanoate metabolism	0.0162	8/151
up	ko00040	Metabolism: Drug metabolism—cytochrome P450	0.0167	8/155
up	ko00071	Metabolism: Pentose and glucuronate interconversions	0.0167	6/88
up	ko00520	Metabolism: Fatty acid degradation	0.0350	6/103
up	ko00030	Metabolism: Amino sugar and nucleotide sugar metabolism	0.0377	4/45
up	ko00983	Metabolism: Pentose phosphate pathway	0.0440	9/228
up	ko00860	Metabolism: Drug metabolism—other enzymes	0.0447	7/149
down	ko00061	Metabolism: Porphyrin and chlorophyll metabolism	0.0020	5/87
down	ko04910	Metabolism: Fatty acid biosynthesis	0.0115	6/247
down	ko01100	Organismal Systems: Insulin signaling pathway	0.0115	20/2498
down	ko01212	Metabolism: Metabolic pathways	0.0115	5/176
down	ko04614	Metabolism: Fatty acid metabolism	0.0115	3/43
down	ko04152	Organismal Systems: Renin-angiotensin system	0.0137	5/191
down	ko04640	Environmental Information Processing: AMPK signaling pathway	0.0284	2/18

## Data Availability

The raw reads of the transcriptome sequencing were deposited in NCBI under the project of PRJNA956151 (SRR24183495–SRR24183499).

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
