# Peer review of "Effect of Ephestia kuehniella Eggs on Development and Transcriptome of the Ladybird Beetle Propylea japonica"

_insects, 2024, doi:10.3390/insects15060407_

Round 1

Reviewer 1 Report

Comments and Suggestions for Authors

The effects of several factitious prey or artificial diets on the survival and growth of ladybird Propylea japonica were compared in this paper. The author want to assess the advantages of E. kuehniella eggs as dietary sources for ladybirds by examining transcriptional regulation. The effect of feeding Pea aphids Megoura crassicauda was similar to that of feeding E. kuehniella eggs without significant difference, but the two treatments were selected in this paper for transcriptome sequencing and comparison. It is highly probable that the differentially expressed genes found are not the main factors affecting the growth of Propylea japonica. This article is not suitable for publication in Insects at the current level. At the same time, the following suggestions are given for reference:

1. The error analysis and significance difference were required for data analysis in Figure 1A. DIET01 and DIET02 both made three duplicates, and the data of the three duplicates were all placed on the graph, which was incorrect for data processing. There were different numbers of repeated worms of DIET01 and DIET02 , and the difference was large, so the experimental design was unreasonable.

2. It is suggested that data in Figure 1B be analyzed for significant differences. Does Figure 1B duplicate Figure 1C and Figure 1D

3. There were only 2 replicates in DIET03, DIET07 and DIET10, while there was only one replicate in DIET05, DIET06, DIET08, DIET11, DIET12, DIET13 and DIET14. This experimental result may be accidental and not statistically significant, and it needs to be repeated several times.

4. In this paper, DIET01 and DIET02 were selected for transcriptome analysis, with DIET01 as the control group and DIET02 as the treatment group. As shown in Figure 1, after feeding DIET01 and DIET02, there was a good survival rate of Propylea japonica, and there was no significant difference in development time and adult body weight. Why these two treatments were chosen to measure the transcriptome and compare.

According to the current experimental results, the survival rate of Propylea japonica treated with DIET01, DIET02 and DIET03 is higher, while the survival rate of DIET05, DIET06, DIET07, DIET08 and DIET09 is lower. The transcriptome of DIET01,DIET02, DIET03 should be compared with that of DIET05, DIET06, DIET07, DIET08 and DIET09, then the reason why DIET01,DIET02, DIET03 treatment is suitable for the growth of Propylea japonica could be summarized.

5. What does Individual refer to in Table 1, the number of eggs fed or the number of Propylea japonica per treatment. In Figures 1C and 1D, do each point on the bar graph represent the developmental time and adult weight of each insect in a certain treatment? The corresponding treatments in Figure 1C and Figure 1D do not match the number of adults, nor do they match the Individual number in Table 1.

6. Whether the correlation between the transcriptome samples is high or not, and whether the repetition is good, it is recommended to place the Pearson correlation analysis chart in the subsequent transcriptome articles.

Comments on the Quality of English Language

The level of written English can continue to improve

Author Response

Main Comment: The effects of several factitious prey or artificial diets on the survival and growth of ladybird Propylea japonica were compared in this paper. The author want to assess the advantages of E. kuehniella eggs as dietary sources for ladybirds by examining transcriptional regulation. The effect of feeding Pea aphids Megoura crassicauda was similar to that of feeding E. kuehniella eggs without significant difference, but the two treatments were selected in this paper for transcriptome sequencing and comparison. It is highly probable that the differentially expressed genes found are not the main factors affecting the growth of Propylea japonica. This article is not suitable for publication in Insects at the current level. At the same time, the following suggestions are given for reference:

Response: The similar developmental phenotypes between the two treatments do not imply that there are no differences at the transcriptional level. These transcriptional regulation factors may not be directly related to development but could influence aspects such as nutrition and immunity. Our goal is to identify these factors to determine the advantages and disadvantages of the diets, thereby formulating optimization strategies. Therefore, we believe that selecting these two diet treatments for transcriptome analysis is valuable and supports the purpose of this study. We added this explanation in the introduction: “Moreover, to explore the advantages of E. kuehniella eggs as a diet, we performed transcriptome profiling to identify differentially expressed genes (DEGs) in P. japonica in response to feeding on E. kuehniella eggs.”

Point 1: The error analysis and significance difference were required for data analysis in Figure 1A. DIET01 and DIET02 both made three duplicates, and the data of the three duplicates were all placed on the graph, which was incorrect for data processing. There were different numbers of repeated worms of DIET01 and DIET02, and the difference was large, so the experimental design was unreasonable.

Response: As suggested, we added a statistical analysis of the differences in survival rates as shown in Figure 1A and the related parts of Methods and Results. We changed “replicates” into “batches” to accurately describe the methods. These batches are independently sampled in different time to evaluate the possible batch effect. We added the explanation in the Methods: “To evaluate the possible batch effect of the main treatments APH and FLO, three batches were conducted for each. For the other treatments, one to three batches were conducted.” Over 40 individual replicates in the main treatments aphids and flour moth eggs) were set, and so we believe the result is reliable.

Point 2: It is suggested that data in Figure 1B be analyzed for significant differences. Does Figure 1B duplicate Figure 1C and Figure 1D

Response: To elucidate the distinctive insights provided by Figure 1B, we added an explanation in the Results section: “Figure 2B integrated the parameters of survival rate, developmental time, and adult weight, demonstrating that APH and FLO exhibited superior performance compared to PUP and HON.”

Point 3: There were only 2 replicates in DIET03, DIET07 and DIET10, while there was only one replicate in DIET05, DIET06, DIET08, DIET11, DIET12, DIET13 and DIET14. This experimental result may be accidental and not statistically significant, and it needs to be repeated several times.

Response: As mentioned in Point 1, we set three batches for the main treatments (DIET01 and DIET02 or APH and FLO in the new version) to evaluate the possible batch effect. Other food treatments might not have the same number of batches because their performance was much lower than DIET01 and DIET02. Thus, the necessity of evaluate batch effect for these diet treatments was less critical for screening high-quality food sources.

Point 4: In this paper, DIET01 and DIET02 were selected for transcriptome analysis, with DIET01 as the control group and DIET02 as the treatment group. As shown in Figure 1, after feeding DIET01 and DIET02, there was a good survival rate of Propylea japonica, and there was no significant difference in development time and adult body weight. Why these two treatments were chosen to measure the transcriptome and compare.

Response: Please refer to the Response to the Main Comment.

Point 5: According to the current experimental results, the survival rate of Propylea japonica treated with DIET01, DIET02 and DIET03 is higher, while the survival rate of DIET05, DIET06, DIET07, DIET08 and DIET09 is lower. The transcriptome of DIET01, DIET02, DIET03 should be compared with that of DIET05, DIET06, DIET07, DIET08 and DIET09, then the reason why DIET01, DIET02, DIET03 treatment is suitable for the growth of Propylea japonica could be summarized.

Response: We agree with the reviewer that the comparing transcriptome of other diet treatments is a valuable analysis. It can help explain the deficiencies of other materials as ladybird diet sources. We will consider conducting this research in the future. For this study, we prioritize focusing on the advantages of flour moth eggs as a food source for ladybirds.

Point 6: What does Individual refer to in Table 1, the number of eggs fed or the number of Propylea japonica per treatment. In Figures 1C and 1D, do each point on the bar graph represent the developmental time and adult weight of each insect in a certain treatment? The corresponding treatments in Figure 1C and Figure 1D do not match the number of adults, nor do they match the Individual number in Table 1.

Response: In Table 1, "individual" refers to the number of Propylea japonica individuals per treatment. To make it clearer, we changed the Table title into “Information of the tested diets, batches, number of individuals tested, and dates of the experiments conducted in this study”. In Figures 1C and 1D, each point on the graph represents the developmental time and adult weight of each insect in a certain treatment. The number of points is less than the number of individuals in Table 1 because the developmental survival rate is not 100%, and some individuals died before pupation and were not included in the statistics of adult weight and developmental time.

Point 7: Whether the correlation between the transcriptome samples is high or not, and whether the repetition is good, it is recommended to place the Pearson correlation analysis chart in the subsequent transcriptome articles.

Response: We used the coefficient of determination (r2) to analyze the correlation of gene expression pattern between transcriptomes. We added the r2 data in the new Table 2.

Point 8: The level of written English can continue to improve.

Response: We used both the grammar check feature in Microsoft Word and AI editing to help to improve the English writing.

Reviewer 2 Report

Comments and Suggestions for Authors

The study seems nice, and well-performed, with many treatments, using advanced laboratory and computational methods. It can have a practical output. There could be two kinds of readers: those interested in breeding ladybirds in the lab and those analysing differential gene expression. For the breeders, authors must explain diverse molecular analyses including the graphs 2 and 3. 

Authors must use italics font for names of genera and species. 

The system of marking treatments is difficult to use. Readers will much appreciate if authors build up a system of abbreviations that will allow them to understand the identity of diets without ever repeatedly looking at the table. I suggest three letters characterizing the food and one number to assign replication. E.g. MEG1, MEG2, MEG3, EPH1, API1, HON1...

There is a formula to combine developmental time and body mass to obtain one that enables to simply rank the treatments. (Ungerová D., Kalushkov P., NedvÄ›d O., 2010: Suitability of diverse prey species for development of Harmonia axyridis and the effect of container size. IOBC-WPRS Bulletin 58: 165–174.)

Comments on the Quality of English Language

Only a few unsuitable wording choices such as refrigerator where they mean freezer.

Author Response

Point 1: The study seems nice, and well-performed, with many treatments, using advanced laboratory and computational methods. It can have a practical output. There could be two kinds of readers: those interested in breeding ladybirds in the lab and those analysing differential gene expression. For the breeders, authors must explain diverse molecular analyses including the graphs 2 and 3.

Response: In the legend of Figure 2, we added “Larger absolute values on the X-axis and higher values on the Y-axis indicate greater differential expression of genes”, along with details of the X and Y axes. Additionally, we included more details in the legend of Figure 3 to make it clearer.

Point 2: Authors must use italics font for names of genera and species.

Response: Done.

Point 3: The system of marking treatments is difficult to use. Readers will much appreciate if authors build up a system of abbreviations that will allow them to understand the identity of diets without ever repeatedly looking at the table. I suggest three letters characterizing the food and one number to assign replication. E.g. MEG1, MEG2, MEG3, EPH1, API1, HON1...

Response: We accepted the reviewer’s suggestion and changed the treatment names accordingly.

Point 4: There is a formula to combine developmental time and body mass to obtain one that enables to simply rank the treatments. (Ungerová D., Kalushkov P., NedvÄ›d O., 2010: Suitability of diverse prey species for development of Harmonia axyridis and the effect of container size. IOBC-WPRS Bulletin 58: 165–174.)

Response: The use of Ungerová et al.’s method is an option. But we consider that Figure 2 presents the data more directly, including developmental time (X-axis), body mass (Y-axis), and survival rate (point size). This method can show the comprehensive effects of different diets on ladybird performance. We added the above explanation in the Results section: “Figure 2B integrated the parameters of survival rate, developmental time, and adult weight, demonstrating that APH and FLO exhibited superior performance compared to PUP and HON.”

Response: All the suggested changes in the PDF were accepted.

Point 6: Only a few unsuitable wording choices such as refrigerator where they mean freezer.

Response: We changed these words as suggested.

Reviewer 3 Report

Comments and Suggestions for Authors

This is a well designed study that examines the effects of several diets on the development and survival of the ladybird beetle Propylea japonica.  This study also includes a transcriptome analysis of the effects of diet on this predatory species, which is an important contribution to understand the underlying mechanisms involved in using diets in biological control programs.  The paper is well written,  I have made several minor suggestions that the authors may wish to consider.

line 84 - delete "daily"

Might add information about when the adults were weighed, for example, on the day of exclusion

lines 86-87 define a biological replicate

lines 128-129  could variation in adult weights among replicates be related to the number of females and males in each replicate.  Females tend to weigh more than males.

In text -- might refer to M. crassicauda (pea aphids) to avoid confusion with the common name pea aphids -- in North America = A. possum

Author Response

This is a well designed study that examines the effects of several diets on the development and survival of the ladybird beetle Propylea japonica. This study also includes a transcriptome analysis of the effects of diet on this predatory species, which is an important contribution to understand the underlying mechanisms involved in using diets in biological control programs. The paper is well written, I have made several minor suggestions that the authors may wish to consider.

Point 1: line 84 - delete "daily"

Response: Done.

Point 2: Might add information about when the adults were weighed, for example, on the day of exclusion

Response: Done. We added this information.

Point 4: lines 86-87 define a biological replicate

Response: We changed “replicates” into “batches” to accurately describe the methods. These batches are independently sampled in different time to evaluate the possible batch effect. We added the explanation in the Methods: “To evaluate the possible batch effect of the main treatments APH and FLO, three batches were conducted for each. For the other treatments, one to three batches were conducted.”

Point 5: lines 128-129 could variation in adult weights among replicates be related to the number of females and males in each replicate. Females tend to weigh more than males.

Response: We did not distinguish between male and female weights in our measurements, which is a limitation of this study. Since normally the adult ratio does not change greatly, we believe that the effect of variation in male / female weight is minor.

Point 6: In text -- might refer to M. crassicauda (pea aphids) to avoid confusion with the common name pea aphids -- in North America = A. possum

Response: We have made this change to "the pea aphids M. crassicauda."

Reviewer 4 Report

Comments and Suggestions for Authors

Manuscript ID: insects-2995842 entitled “Effect of Ephestia kuehniella eggs on development and transcriptome of the ladybird beetle Propylea japonica” by Guannan Li, Pei-Tao Chen, Mei-Lan Chen, Tuo-Yan Chen, Yu-Hao Huang, Xin Lü, Hao-Sen Li, and Hong Pang submitted to Insects,  Genetics and Evolution of Ladybird Beetles in Biological Control, is a well written article that should be considered for publication pending minor revision. I added some editorial notes to the attached PDF for authors’ revision, some of which are:  

·         Full scientific name should be italic documented with authors, order, and family when first mentioned in the text throughout the manuscript and references section.

·         Add statistical values (F. df, P) when appropriate.   

·         Use (–) not (-) hyphen throughout the refences section and bold the year.

Author Response

Point 1: Manuscript ID: insects-2995842 entitled “Effect of Ephestia kuehniella eggs on development and transcriptome of the ladybird beetle Propylea japonica” by Guannan Li, Pei-Tao Chen, Mei-Lan Chen, Tuo-Yan Chen, Yu-Hao Huang, Xin Lü, Hao-Sen Li, and Hong Pang submitted to Insects, Genetics and Evolution of Ladybird Beetles in Biological Control, is a well written article that should be considered for publication pending minor revision. I added some editorial notes to the attached PDF for authors’ revision, some of which are:  

Response: The GPS coordinates of the collection sites was added.

Point 2: Full scientific name should be italic documented with authors, order, and family when first mentioned in the text throughout the manuscript and references section.

Response: Full scientific name The full scientific names and taxonomic information of ladybirds, moths and aphids were added.

Point 3: Add statistical values (F. df, P) when appropriate.   

Response: Done.

Point 4: Use (–) not (-) hyphen throughout the refences section and bold the year.

Response: All references have been corrected accordingly.

Response: All the changes suggested in the PDF were accepted.

Round 2

Reviewer 1 Report

Comments and Suggestions for Authors

I still think this article is not suitable for publication in Insects at the current level. The author did not do a good job of addressing the issues I raised and correcting the errors in the article.

1. “Transcriptome profiling of larvae raised on E. kuehniella eggs and M. crassicauda revealed that genes upregulated in the former group were enriched in metabolic pathways associated with carbohydrates, lipids, and other essential nutrients, indicating a higher nutrient content in E. kuehniella eggs compared to natural prey. “ is not correct. It is not the up-regulation of genes on the nutritional signaling pathway that indicates nutritional abundance.

2. According to transcriptome analysis of these two diet treatments of feeding Pea aphids Megoura crassicauda or feeding E. kuehniella eggs, we can't draw that conclusion “the effectiveness of E. kuehniella eggs as a food source is due to their appropriate nutrients and lower microbial challenge for the ladybirds.” Because Pea aphids Megoura crassicauda is also a good feed。

3. The 3 replicates of a treatment should be put together for statistical analysis of the differences, and there would be an error bar for differences between 3 replicates. According to the author's description, Fig1A data of APH or FLO is the result of 3 batches of experiments, each batch has only one theoretical experimental result. How does the error bar come from? I suggest the author learn about data analysis.

4. Repeated validation cannot be omitted just because the results of one experiment are not good. An experiment is likely to happen by chance. The workload of the experiment is also insufficient.

5.  In Figures 1C and 1D, each point on the bar graph represent the developmental time and adult weight of each insect in a certain treatment. However, the number of individuals corresponding to Fig1C and Fig1D is also inconsistent.

Comments on the Quality of English Language

English writing needs to be improved

Author Response

Point-by-point response

Main Comment: I still think this article is not suitable for publication in Insects at the current level. The author did not do a good job of addressing the issues I raised and correcting the errors in the article.

Response: Thank you for your valuable comments on our manuscript. We have carefully reviewed the two reports and revised the manuscript accordingly. Specifically, we have:

  1. Updated the statements regarding the relationship between gene expression and diet content.
  2. Added statistical analyses to test the differences in life history traits between the aphid and flour moth egg treatments.
  3. Justified the rationale behind our experimental settings.
  4. Improved the English writing throughout the manuscript.

We appreciate your insightful feedback and believe these revisions enhance the clarity and robustness of our work.

Point 1: “Transcriptome profiling of larvae raised on E. kuehniella eggs and M. crassicauda revealed that genes upregulated in the former group were enriched in metabolic pathways associated with carbohydrates, lipids, and other essential nutrients, indicating a higher nutrient content in E. kuehniella eggs compared to natural prey.” is not correct. It is not the up-regulation of genes on the nutritional signaling pathway that indicates nutritional abundance.

Response: Some studies have suggested that differentially expressed metabolism-related genes in insects respond to the varied nutritional compositions of their diets (as discussed in Lines 192-205). However, we did not test the nutritional content differences between aphids and flour moth eggs. Additionally, we cannot rule out the possibility that other factors also con-tributed to the differential expression of these genes. Therefore, we revised our statement to “This suggests that E. kuehniella eggs may have a higher nutrient content compared to natural prey.” We also added the above explanation to the Discussion section.

Point 2: According to transcriptome analysis of these two diet treatments of feeding Pea aphids Megoura crassicauda or feeding E. kuehniella eggs, we can't draw that conclusion “the effectiveness of E. kuehniella eggs as a food source is due to their appropriate nutrients and lower microbial challenge for the ladybirds.” Because Pea aphids Megoura crassicauda is also a good feed。

Response: We agree that both pea aphids Megoura crassicauda and E. kuehniella eggs are suitable diets for the ladybird Propylea japonica based on the comparison of life history traits. To make this point clear, we revised this sentence to: “We suggest that the difference between E. kuehniella eggs and M. crassicauda as food sources for P. japonica lies in their nutrient and microbial contents.” We also made this change in the Discussion section. We also changed the related Introduction sentence into “We also performed transcriptome profiling to identify differentially expressed genes (DEGs) in P. japonica in response to feeding on E. kuehniella eggs.”

Point 3: The 3 replicates of a treatment should be put together for statistical analysis of the differences, and there would be an error bar for differences between 3 replicates. According to the author's description, Fig1A data of APH or FLO is the result of 3 batches of experiments, each batch has only one theoretical experimental result. How does the error bar come from? I suggest the author learn about data analysis.

Response: In the previous version, we calculated the means and standard errors of each batch and used a non-parametric method to compare the difference in means. In this version, following the reviewer's suggestion, we also calculated the means and standard errors of each treatment. We added the related descriptions in the Methods and Results sections: we included “Statistical analyses for comparing life history traits among different treatments and batches were carried out using R software” in Line 111 and “None of the life history traits were significantly different between the APH and FLO treatments (Table 2)” in Line 148. The results are now presented in the new Table 2.

Point 4: Repeated validation cannot be omitted just because the results of one experiment are not good. An experiment is likely to happen by chance. The workload of the experiment is also insufficient.

Response: We did not omit to validate the results. We understand that variation among individuals and batches may impact the outcomes. Therefore, for the two main treatments, we included over 40 individual replicates per batch and conducted three batch replicates per treatment. We believe that this experimental setup is sufficient to support the results.

Point 5: In Figures 1C and 1D, each point on the bar graph represent the developmental time and adult weight of each insect in a certain treatment. However, the number of individuals corresponding to Fig1C and Fig1D is also inconsistent.

Response: The difference arises because Figure 1C displays all individuals that reached adulthood, whereas Figure 1D specifically displays the weight of female adults. For instance, APH02 includes 24 adult individuals in Figure 1C and only 12 female adult individuals in Figure 1D. We changed the Figure legend accordingly, and also uploaded the raw data of life history traits in case it is needed:

https://www.dropbox.com/scl/fi/stl0dyox0s2mwfuu4uaan/life_history_rawdata.xlsx?rlkey=0ptxdyeqprqs2o7vfp7v52c2d&st=xv3pgkis&dl=0

Comments on the Quality of English Language: English writing needs to be improved.

Response: The English writing was checked by both AI and a colleague fluent in English writing. Consequently, we made several adjustments. For example:

Line 13: changed into “Improving augmentative biological control relies on development of a cost-effective and readily available diet for rearing natural enemy insects.”

Line 72: changed into “This study aimed to assess the suitability of E. kuehniella eggs as a diet for P. japonica and explore the differences between E. kuehniella eggs and natural prey as food sources for this ladybird.”

Line 186: changed into “Although there was variability in performance among batches in our experiments, the life history traits of P. japonica reared on both E. kuehniella eggs and aphids showed relatively similar patterns.”